Using cloud-based mobile technology for assessment of competencies among medical students

Ferenchick Gary S. Gary.ferenchick@ht.msu.edu
Solomon David
Department of Internal Medicine, Division of General Medicine, Michigan State University , East Lansing, MI , USA
Hochheiser Harry
Electronic publication date: 2013 Sep 17
Publication date: 2013
Volume: 1
Electronic Location ID: e164
Received 2013 May 26; Accepted 2013 Aug 29
Copyright: © 2013 Ferenchick and Solomon
Copyright year: 2013
Copyright holder: Ferenchick and Solomon
License: This is an open access article distributed under the terms of the Creative Commons Attribution License, which permits unrestricted use, distribution, and reproduction in any medium, provided the original author and source are credited.
License URL: https://creativecommons.org/licenses/by/3.0/

Keywords: Educational technology, Educational measurement, Medical students, Smart phones, Competency based assessment, Direct observation, Medical faculty, Clinical competence, iPhone, miniCEX

Funding: Division of Medicine, Bureau of Health Professions, Health Services and Resources Administration D56HP05217 This study was funded in part by a grant D56HP05217 from the Division of Medicine, Bureau of Health Professions, Health Services and Resources Administration. The funders had no role in study design, data collection and analysis, decision to publish, or preparation of the manuscript.

==============================
Valid, direct observation of medical student competency in clinical settings remains challenging and limits the opportunity to promote performance-based student advancement. The rationale for direct observation is to ascertain that students have acquired the core clinical competencies needed to care for patients. Too often student observation results in highly variable evaluations which are skewed by factors other than the student’s actual performance. Among the barriers to effective direct observation and assessment include the lack of effective tools and strategies for assuring that transparent standards are used for judging clinical competency in authentic clinical settings. We developed a web-based content management system under the name, Just in Time Medicine (JIT), to address many of these issues. The goals of JIT were fourfold: First, to create a self-service interface allowing faculty with average computing skills to author customizable content and criterion-based assessment tools displayable on internet enabled devices, including mobile devices; second, to create an assessment and feedback tool capable of capturing learner progress related to hundreds of clinical skills; third, to enable easy access and utilization of these tools by faculty for learner assessment in authentic clinical settings as a means of just in time faculty development; fourth, to create a permanent record of the trainees’ observed skills useful for both learner and program evaluation. From July 2010 through October 2012, we implemented a JIT enabled clinical evaluation exercise (CEX) among 367 third year internal medicine students. Observers (attending physicians and residents) performed CEX assessments using JIT to guide and document their observations, record their time observing and providing feedback to the students, and their overall satisfaction. Inter-rater reliability and validity were assessed with 17 observers who viewed six videotaped student-patient encounters and by measuring the correlation between student CEX scores and their scores on subsequent standardized-patient OSCE exams. A total of 3567 CEXs were completed by 516 observers. The average number of evaluations per student was 9.7 (±1.8 SD) and the average number of CEXs completed per observer was 6.9 (±15.8 SD). Observers spent less than 10 min on 43–50% of the CEXs and 68.6% on feedback sessions. A majority of observers (92%) reported satisfaction with the CEX. Inter-rater reliability was measured at 0.69 among all observers viewing the videotapes and these ratings adequately discriminated competent from non-competent performance. The measured CEX grades correlated with subsequent student performance on an end-of-year OSCE. We conclude that the use of JIT is feasible in capturing discrete clinical performance data with a high degree of user satisfaction. Our embedded checklists had adequate inter-rater reliability and concurrent and predictive validity.

Introduction

The assessment of the clinical competence of a medical student is challenging. A competency is, “…an observable ability of a health professional related to a specific activity that integrates knowledge, skills, values, and attitudes. Since they are observable, they can be measured and assessed”. Although seemingly straightforward, competency based education is of limited usefulness in guiding the design and implementation of educational experiences if they are not tied to specific learning objectives (Whitcomb, 2004). Additionally, learning objectives are of limited usefulness if they are not available to students and faculty when interacting with patients. Finally, observation and assessment help neither students nor patients if they are not captured and documented in a way that facilitates learner specific plans for improvement and excellence. We present a generalizable initiative that makes national curricula functional in local learning environments and improves, and simplifies, observation based assessments and performance-based data tracking for faculty and learners.

Materials & Methods

Content manager

We developed a mobile, cloud-based application called Just in Time Medicine (JIT) that functions effectively on smart phones, tablets and laptop computers. The mobile application is supported by a self-service web-based content management system designed with the explicit aim of enabling users with average computing skills to build their own customizable content, including criterion-based checklists that can then be delivered to any internet enabled device such as a smart phone or tablet.

For this project, we utilized nineteen core training problems from the nationally validated Clerkship Directors in Internal Medicine (CDIM) curriculum and combined these training problems with the observable competencies of communication skills, history taking and physical examination to create problem and task specific checklists. For each assessment, the software calculates the students’ performance by determining the percentage of all potential items performed correctly, and an algorithm generated grade of “not done/unsatisfactory”, “needs improvement” or “well done” is calculated depending on the percentage of items performed correctly. In general, if a student achieved 80% of the expected items correctly they received a “well done” grade; performing 30–79% of the expected items resulted in a “needs improvement” grade, and <30% an “unsatisfactory”. Figures 1 and 2 present screen shots for the process of building checklists using our content manager for the problem altered mental status and the competency history taking. Additionally, Figs. 3 and 4 show how the assessment tools are displayed on the user’s device. Figures 5–7 show the permanent cloud-based reporting options associated with individual assessments. A fully functional version of JIT can be accessed at: www.justintimemedicine.com/mobile; log in username is testuser@journal.com, and the password is test. To access examples of Cloud-based performance reporting, go to www.justintimemedicine.com; username: testadministrator@journal.com and password: test.

Figure 1 Step 1. Content manager for development of assessment tools.

Using simple interfaces, faculty adds content (e.g., the problem altered mental status) and the specific competency to be assessed (e.g., history taking).

Figure 2 Step 2. Content manager for development of assessment tools.

Using the self-service web-based content management system, faculty then adds assessment questions reflecting specific criterion-based outcomes (e.g., The student started the interview with open-ended questions).

Figure 3 Criterion-based assessment for altered mental status and history-taking as displayed on the mobile device for use anytime and anywhere.

A displays how the specific checklist is accessed on the device; B displays the criterion-based tasks, which are defaulted to No and change to Yes (C) once the task is completed by the learner. D displays the algorithm generated grade.

Figure 4 Evaluator information is collected using simple interfaces on the device after the assessment is completed, including open-ended qualitative comments.

Faculty enters information concerning their observation (A), and their feedback and action plans (B). A color coded competency registry is displayed on the learner’s device (C). Note in B, the evaluator has the option to have an email link sent to him/her to complete the qualitative assessment at a later time. All evaluations become part of the learner’s cloud-based permanent record.

Figure 5 Detailed cloud-based reporting options.

One of the web-based permanent records of the students’ performance; displaying the item(s) assessed, the percentage of potential items correctly performed, and algorithm generated grade and evaluators written comments on the learners performance (note all of these features are editable, based upon the users’ needs).

Figure 6 JIT detailed cloud-based reporting options.

With the click of a hyperlink, a detailed list of all the items that were either performed or not by the student is displayed.

Figure 7 JIT cloud-based reporting options.

Another option for a cloud-based record or registry of the learner’s performance. This image represents a milestone based report with the identified milestones (A) the milestone subcompetencies (B) a color-coded table of all of the learners assessments (C) a roll-over option (D) identifies which specific assessment is represented in each cell. This table shows the ACGME competency taxonomy for internal medicine.

Goals and hypotheses

In introducing JIT in our clerkship, we hypothesized that JIT would: (1) facilitate the direct observation and provision of feedback to trainees on their clinical competencies; (2) generally be accepted by faculty; (3) provide a means for recording the observations of trainee performance, and (4) possess adequate reliability and validity.

Setting

The College of Human Medicine (CHM) at Michigan State University is a community-based medical school with clinical training in 7 communities throughout Michigan. Between July 2010 and October 2012 we implemented JIT as an integral part of the Internal Medicine Clerkship among 367 students. Each student was required to complete ten directly observed clinical evaluation exercises (CEXs) with real patients in authentic clinical settings. A CEX is a short (generally <20 min) directly observed trainee – patient interaction (e.g., history-taking, examination, counseling, etc.); faculty observes, rates, and provides written comments on the interaction. Students received an orientation to the CEX application and were required to become familiar with the software. Evaluators (attending faculty and residents) received an email on the importance of direct observation and the basic functionality of the CEX application.

In general, students chose the patient, problem and competency upon which to be assessed. At the time of the assessment, students handed their mobile device, with the checklists displayed, for evaluator use during the assessed interaction. A total of 516 evaluators subsequently used JIT to guide their observations and assessments of students interacting with patients.

Data collection

We collected the following data: the specific training problems and competencies observed and assessed by the evaluators, the grades associated with the observation and descriptive data from faculty on the use of JIT. Descriptive data was collected from the faculty via “pull-down” menus located on the last screen of each assessment. A screen shot of the interface is displayed in Fig. 4.

Reliability and validity assessments

A group of 17 evaluators, 9 internal medicine residents and 8 general internist faculty members viewed and rated six scripted videotaped encounters using JIT. Each case was scripted for both satisfactory and unsatisfactory performance. These cases have been previously validated by Holmboe as representing levels of competence which range from unequivocally poor to satisfactory (Holmboe, Hawkins & Huot, 2004). The sample of raters reflected the number we could reasonably obtain given our small general internal medicine faculty and residency program. We felt it was adequate to provide a stable estimate of the inter-rater reliability of the assessment process. We calculated the inter-rater reliability using a formula developed by Ebel and implemented using software developed by one of the authors (Ebel, 1951; Solomon, 2004). All other statistical analyses were performed with SPSS version 21.

Results

Number and types of evaluations

Five hundred sixteen evaluators used the application to assess 367 students for a total of 3567 separate assessments. The number of CEX’s completed per student was 9.7 (±1.8) and the average number of CEX’s completed per faculty was 6.9 (±15.8). The average number of training problems a student was assessed on was 6.7; of the three competency domains of communication skills, history taking, and physical examination 68% of the students had at least one evaluation in each of the three categories.

In terms of the grades, time variables and satisfaction, ∼83% of the encounters were associated with a “well done” grade, and on average students were credited with performing ∼86% of the items correctly (Fig. 8). Between 43–50% of the CEX assessments took <10 min as estimated by the faculty, and in ∼69% of the encounters feedback was estimated to occur in less than 10 min. In 92% of the encounters, faculty rated that they were either satisfied or highly satisfied with the CEX.

Figure 8 Bar chart of grade distribution comparing resident to faculty raters.

The estimated inter-rater reliability of a single rater observing the videotaped encounters was 0.69 (slightly higher for faculty at 0.74 vs. residents at .64). In judging the same simulated patient case scripted to be satisfactory and non-satisfactory, the residents and faculty using JIT discriminated between the satisfactory and non-satisfactory performance. The mean number of items checked for the videotapes scripted for unsatisfactory performance was 35% vs. 59% for those scripted for more satisfactory performance. We believe this provides evidence supporting the construct validity of JIT.

To assess predictive validity, we calculated a Pearson product moment correlation between “gateway” performance assessment examinations taken by 282 students at the end of their third year required clerkships with the CEX assessments obtained by JIT. There was a small (but statistically significant 0.144, p = .008) correlation between students’ CEX scores and communications skills in the gateway performance assessment exam.

Discussion

Although national learning objectives have been published for all core clerkships, their usefulness for assessing learning outcomes has been limited. As an example, the core competency gathering essential and accurate information seems relatively straightforward. However, when applied to a single condition such as chronic obstructive pulmonary disease, there are at least 28 specified clinical tasks related to history taking and performing a physical examination that a student should demonstrate to meet the expected outcomes as defined in the Clerkship Directors in Internal Medicine (CDIM) curricular objectives for that problem. Of these 28, how many will a faculty evaluator remember when assessing the student? More importantly how many can they remember and what level of consistency will there be among preceptors providing feedback to students?

If we take almost any clinical skill and start to dissect it, we find very quickly that existing human memory is insufficient in recalling all of the explicit steps related to potentially hundreds of conditions that help frame the expected outcomes of a trainee’s educational experience and curricula. As the expectations for assessment of discrete competencies increases, the evaluation burden for educators, students and administrators becomes progressively more educationally incomplete and logistically unmanageable.

The inability of faculty to remember and accurately assess for outcomes related to potentially hundreds of discrete educational objectives while evaluating trainees in clinical settings is one of the major reasons faculty have a hard time reliably discriminating unsatisfactory from satisfactory performance, as has been noted by many authors over the past decade using paper-based systems (Holmboe, Hawkins & Huot, 2004; Kogan et al., 2011). For example, in a study of mini-CEX evaluations among 300 medical students, Hill noted that problems existed, “in trying to ensure that everyone was working to the same or similar standards” (Hill et al., 2009). In another study of 400 mini-CEX assessments, Fernando concluded faculty evaluators were unsure of the level of performance expected of the learners (Fernando et al., 2008). Hasnain noted that poor agreement among faculty evaluating medical students on a Family Medicine Clerkship was due to the fact that “Standards for judging clinical competence were not explicit” (Hasnain et al., 2004). In a randomized trial of a faculty development effort, Holmboe studied the accuracy of faculty ratings by having them view videotaped trainee-patient encounters that were scripted to portray three levels of proficiency; unsatisfactory, marginal or satisfactory. Faculty viewing the exact same encounter varied widely in their assessment of trainee competence, with ratings from unequivocally unsatisfactory (CEX scores of scores 1–3) to unequivocally superior (CEX scores of 7–9), regardless of whether the video was scripted to be unsatisfactory or not. After an intensive 4 day faculty development workshop in which participants were tasked with developing a shared mental model of what specific competencies should look like, problems still existed among faculty in discriminating satisfactory from unsatisfactory performance in these scripted encounters (Holmboe, Hawkins & Huot, 2004).

Kogan noted that in the absence of easily accessible frameworks, faculty evaluators default back to a myriad of highly variable evaluation strategies including such idiosyncratic features as instinct, “gut feelings”, “unsubstantiated assumptions” and the faculty members’ emotional response to providing feedback. What she also noted was that faculty raters commonly fail to use existing frameworks or external standards in guiding their evaluations of trainees, thus explaining much of the well-recognized problems with poor validity and inter-rater reliability associated with clinical evaluations (Kogan et al., 2011).

Given these realities, it is not surprising that medical trainees commonly do not view the feedback received from faculty as credible nor influential in learning, especially if the feedback was not immediate and tied to the trainees’ clinical work-place performance (Watling et al., 2012). Enhancing the effectiveness of clinical assessments, the delivery of feedback related to learning objectives and the creation of better systems for documenting faculty observations are commonly cited needs in medical education (Hasnain et al., 2004; Howley & Wilson, 2004; Torre et al., 2007; Hauer & Kogan, 2012; Whitcomb, 2002).

Given these and other trends, systems that are capable of disseminating curricular objectives to students and faculty and which also enable criterion-based assessment have become key educational needs. We believe that cloud-based technology, appropriately applied to maximize efficiency, can contribute to optimizing the learning environment by directly aligning learning objectives from national disciplinary curricula with assessment tools for use by students and faculty anywhere and anytime, especially at the bedside.

In our first feasibility study, we demonstrated our ability to deliver national educational objectives published by the CDIM to electronic hand-held personal digital assistants (PDAs) such as Palm® and PocketPC® devices (Ferenchick, Fetters & Carse, 2008). In a second feasibility study, we subsequently demonstrated that this system could be used to deliver, and successfully implement, competency-based checklists for student assessment related to the CDIM curricular objectives using PDAs (Ferenchick et al., 2010). Data from these studies helped us determine that the distribution and use of curricular objectives and related assessment tools by students and faculty in our geographically dispersed medical school could be facilitated with just in time mobile technology. Importantly, we also determined that students and preceptors valued the fact that the content and expected competencies were transparent and such transparency facilitated learner assessment (Ferenchick et al., 2010). However, technical issues with PDAs – such as lack of direct internet connection and the requirement to “synchronize” data from PDAs to the web using desktop computers – limited the practicality of PDA based assessment; a process that is not needed with contemporary internet enabled devices such as iPads, iPhones and other smartphones. These devices have become almost ubiquitous in the past four years and we have leveraged this trend to evolve JIT to a platform-neutral Cloud-based system. The displayed assessment tools function like an “application” on mobile devices, but are device-agnostic in that they function on all internet-enabled devices, including smartphones.

Our study, like most others, has several inherent limitations. First, this is a single institution study and these results may not be generalizable. Future studies should focus on the use of this technology in other settings. Second, establishing the reliability of all of the customized checklists within the CEX application is needed, as is establishing its reliability in real clinical settings such as the hospital wards. Third, we have not established the validity of the electronic grading algorithm. Fourth, like many tools for direct observation, we have not established the effect of this tool on learning nor the transfer of acquired clinical skills to other areas, or the effect that such direct observation has on the most important outcome of patient care.

Conclusions

We have established that just in time Cloud-based mobile technology has great potential in competency-based medical education. Although not an objective of this study, we believe such technology holds great promise for use in authentic clinical settings for measuring student achievement related to educational milestones. Additionally, given the time and cost constraints associated with traditional faculty development efforts, we believe that systems such as JIT have great potential in operationalizing “just in time” faculty development.

Supplemental Information

Supplemental Information 1 IRB determination form

Click here for additional data file.

Supplemental Information 2 Primary data source for this study

Click here for additional data file.

Additional Information and Declarations

Competing Interests

Author Contributions

Human Ethics

Dr David Solomon is an Academic Editor for PeerJ.

Gary S. Ferenchick conceived and designed the experiments, performed the experiments, analyzed the data, wrote the paper.

David Solomon analyzed the data, wrote the paper.

The following information was supplied relating to ethical approvals (i.e., approving body and any reference numbers):

Our medical school has created an “Honest Broker System” for conducting research on student performance data that are collected as a regular part of the students’ educational activities. A designated employee of the medical school, with access to these data, has been designated as the “Honest Broker”. In an honest broker system a person or agency that has access to multiple human subject datasets collected for non-research purposes creates a de-identified dataset that can be used for research purposes without posing risk to the subjects (Boyd et al., 2007). This approach has been used in various types of clinical research – though, to the best of our knowledge, it has not been applied in educational research at other institutions.

At Michigan State University this designated individual created the dataset used in this study and made it available to our research team after removing all identifiers. The Social Science/Behavioral/Education Institutional Review Board (SIRB) at Michigan State University granted our study an exempted review.

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
