# Peer review of "Using cloud-based mobile technology for assessment of competencies among medical students"

_PeerJ, doi:10.7717/peerj.164_

## Round 0.1 · original submission · Major Revisions

The reviewers seem to be in agreement that additional detail regarding the stastical design and analyses would be helpful for understanding the experiment and its results. Both reviewers provided suggestions for additional detail that would make this paper both clearer and more informative. Please consider these comments in your revisions.

Reviewer 1 ·

Basic reporting

The authors have met standards.

Experimental design

The authors hypothesized that JIT would: 1) facilitate the
direct observation and provision of feedback to trainees on their clinical competencies; 2) generally be accepted by faculty; 3) provide a means for recording the observations of trainee performance, and 4) possess adequate reliability and validity.

To evaluate these hypotheses, the authors collected following data: the specific training problems and competencies observed and assessed by the evaluators, the grades associated with the observation and descriptive data from faculty on the use of JIT.

The experimental design appears adequate, however I do wish more qualitative data regarding general acceptance by faculty would have been collected an analyzed.

The explanation of the statistical methods is inadequate and fails to meet standards. For instance, the authors fail to describe adequate methods regarding the statistical computation of inter-rater reliability. This outcome can be assessed through a number of statistical tests, and the specific method should be described. In addition, correlation coefficient was computed and presented, but the method was not described (e.g. Spearman's or Pearson's).

Validity of the findings

The findings generally support determination of the four hypotheses described by the authors. I would like to see more data to support hypothesis #2 (acceptance by faculty) and a more thorough description of the statistical methodology in order to assess the conclusion of hypothesis #4 (possess adequate reliability and validity).

It would be nice for the authors to identify one primary outcome measure for the study (which of the four stated hypotheses was the major focus of this study?).

It appears a very specific number (17) of evaluators were selected to test for reliability and validity. How did the authors arrive at this number of evaluators? Did they compute a sample size based on expected reliability and minimum reliability? What was the method by which they computed the sample size of 17 evaluators for the reliability testing? Was the group of 17 evaluators a convenience sample or were they a specific target group? What was the demographic composition of the group (it appears some residents were used as evaluators, but the number is never mentioned). There does not appear to be sufficient operational detail regarding the specifics of this aspect of the study for a reader to reproduce the results.

Additional comments

This paper reports the use of a tool for enabling educators to assess medical student performance at the point of care using a mobile-responsive web-based application. While the application itself is not novel, the implementation and execution of the concept appears elegant.

Some areas for future work: 1) integration into institution-centric learning management systems, 2) incorporation of reflective learning opportunities for students to self-assess their own performance, 3) ability to provide audio or video-based feedback for learners using the mobile interface, 4) ability for evaluators to use their own devices for logging feedback rather than using the student's device.

A qualitative analysis of user experience and feedback from learners about this system would be useful to gauge difficulties with the system and/or ways to scale implementation of this tool widely throughout the institution.

The manuscript should be revised to provide a more detailed description of the statistical methods used in the paper. More details about the reliability and validity testing should also be included.

Reviewer 2 ·

Basic reporting

The authors describe a cloud-based application for assessment of competencies in medical students. The application was implemented and feasibility evaluations were done.

In the Introduction, provide some background description of CEX with some references since readers may not be familiar with CEX. An illustrative example of a CEX will also be useful.

Minor points:
Use lower case “c” in Cloud-based
Use “application” instead of the abbreviation “app”

Experimental design

In the Materials & Methods section:

Mention for what platforms the application is available (iOS, Android, etc.)

What cut-offs were used to determine the three grades that were generated by the algorithm?

Give details about how CEX assessment time was measured.

Give details of the instrument/questions that were used to assess the faculty’s satisfaction with the app.

Provide the statistic that was used to measure inter-rater reliability and the statistical software used to do the analysis.

Provide details of the statistic and the statistical software used for analyzing the correlation between the “gateway” performance assessment examinations and the CEX assessments.

Validity of the findings

In the Results section:

It will be useful to provide more detailed results of grades, time and satisfaction for the 3567 assessments. Bar charts for each of the 3 variables that show the percent breakdown will be useful.

For interrater reliability provide 2-sided confidence intervals along with the statistic for faculty alone, residents alone and faculty and residents combined.

Additional comments

Are there tools (e.g. paper based instruments) that have been described in the literature for assisting in CEX assessments? If so, it would be useful to include brief descriptions of them in the Discussion section.

---

## Round 0.2 · accepted · Accept

Thank you for your prompt responses to the reviewers' comments.

Reviewer 1 ·

Basic reporting

The authors have addressed my previous comments in a satisfactory manner.

Experimental design

The authors have addressed my previous comments in a satisfactory manner.

Validity of the findings

The authors have addressed my previous comments in a satisfactory manner.

Additional comments

The authors have addressed my previous comments in a satisfactory manner.